# Cost-Effective High-Performance Concrete: Experimental Analysis on Shrinkage

**DOI:** 10.3390/ma12172730

**Published:** 2019-08-26

**Authors:** Barbara Kucharczyková, Dalibor Kocáb, Petr Daněk, Ivailo Terzijski

**Affiliations:** Faculty of Civil Engineering, Brno University of Technology, Veveří 331/95, 602 00 Brno, Czech Republic

**Keywords:** high-performance concrete, shrinkage, mass losses, autogenous, strain gauge, shrinkage molds

## Abstract

This paper focuses on the experimental determination of the shrinkage process in Self-Compacting High-Performance Concrete (SCC HPC) exposed to dry air and autogenous conditions. Special molds with dimensions of 100 mm × 60 mm × 1000 mm and 50 mm × 50 mm × 300 mm equipped with one movable head are used for the measurement. The main aim of this study is to compare the shrinkage curves of SCC HPC, which were obtained by using different measurement devices and for specimens of different sizes. In addition, two different times *t*_0_ are considered for the data evaluation to investigate the influence of this factor on the absolute value of shrinkage. In the first case, *t*_0_ is the time of the start of measurement, in the second case, *t*_0_ is the setting time. The early-shrinkage (48 h) is continuously measured using inductive sensors leant against the movable head and with strain gauges embedded inside the test specimen. To monitor the long term shrinkage, the specimens are equipped with special markers, embedded into the specimens’ upper surface or ends. These markers serve as measurement bases for the measurement using mechanical strain gauges. The test specimens are demolded after 48 h and the long term shrinkage is monitored using the embedded strain gauges (inside the specimens) and mechanical strain gauges that are placed, in regular intervals, onto the markers embedded into the specimens’ surface or ends. The results show that both types of measurement equipment give a similar result in the case of early age measurement, especially for the specimens cured under autogenous conditions. However, the early age and especially long term measurement are influenced by the position of the measurement sensors, particularly in the case of specimens cured under dry air conditions. It was proven that the time *t*_0_ have a fundamental influence on the final values of the shrinkage of investigated SCC HPC and have a significant impact on the conclusions on the size effect.

## 1. Introduction

The definition of High-Performance Concrete (HPC) and Ultra-High-Performance Concrete (UHPC) is developing along with the development of the properties of these materials. It has been shown that in both cases the compressive strength is only one of the parameters which determine the performance of the concrete and concrete members, which has changed completely the approach to the design of the concrete composition [1]. The subsequent development of the HPC resulting from the rising demands for the strength and durability of the concrete structures led to the establishment of the new term, UHPC, with the aim to distinguish these two materials. There are no unified standards for the design of composition of both HPC and UHPC throughout the world because of various raw materials available in particular countries, which lead to the individual design of HPC and UHPC for the specific application. This fact is one of the factors which affect the costs of the construction.

The latest development of UHPC trends in the design of a cost-effective UHPC, which combines the demand for the high-performance properties of the material with the cost of construction [2,3]. This approach brings new challenges for concrete technologists, who have to use locally available raw materials for production of UHPC with excellent properties. To reduce the material costs of UHPC, quartz aggregate is replaced by natural aggregate or different types of reactive by-products are used to decrease the cement consumption. The demands on the properties of cost-effective HPC and UHPC must be clearly defined before the start of the design of the concrete composition to achieve a high-performance material using raw materials of lower quality. The authors of Reference [2] focus on the optimization and performance of cost-effective UHPC and propose a mix design method for UHPC prepared with high-volume supplementary materials and conventional concrete sand, including steel fibers. Reference [3] deals with the development of cost-effective UHPC for Colorado’s infrastructure using different silica compounds including steel and polypropylene fibers.

In some cases, due to the low quality of the raw material, not all requirements posed to the UHPC can be met together—high compressive and tensile strength, high durability, low shrinkage, etc. These consequences can be minimized by the usage of one or more “excellent admixtures”, as was applied by the authors of this paper (see Section 2).

The shrinkage process belongs to one of the most problematic properties of HPC and UHPC. Both concretes are under the risk of shrinkage cracking because of the high value of the binder-aggregate ratio (b/a). In addition, the low water-cement ratio (w/c) increases the risk of the existence of the autogenous shrinkage. Due to the different behaviors of Normal Strength Concrete (NSC) and HPC during the early solidification process, the measurement techniques commonly used for the measurement of shrinkage in NSC are not fully suitable for the determination of the shrinkage process in HPC and UHPC. In the case of concretes with the risk of autogenous shrinkage, the measurement has to start as early as possible. This was already reported by Aictin in 1999 [4] and is also recommended by Yoo et al. [5]. Currently, there are two commonly used approaches on how to measure the autogenous strain effectively. The first one is the usage of the corrugated tube method [6] which is limited by the size of aggregate (max. grain size of 4.75 mm). The second one is the use of an appropriate type of strain gauge intended for embedding into the sealed specimens (the low-stiffness type of gauge must be used; its length must be appropriate to the maximum grain size of aggregate). The costs of the strain gauges lead to an increase in the overall costs of the HPC design. There are also different types of shrinkage molds with one or two movable heads equipped with the inductive or contactless sensors available. This way of shrinkage measurement is more cost-effective because the molds are reusable. The laboratory equipment intended for drying shrinkage measurement is mostly designed to start the measurement after the demolding of the test specimens, typically after 24 h of aging. The embedded strain gauges of various types are also used for laboratory as well as for in-situ measurements.

It is well known that different measurement techniques give different shrinkage curves. This is due to the different constructions and operating principles of the particular measurement devices which lead to the different start of measurement. Another reason is the different time when the evaluation of the measured data starts. Most often the data are zeroed at the time of the start of measurement, at the setting time, at the time of reaching the peak temperature or at the time of demolding of the test specimens. There are no references available where the complex comparison of the measurements techniques could be found.

The paper focuses on the determination of the overall process of shrinkage using specially modified shrinkage molds with one movable head equipped with an inductive sensor. The vibrating wire-type strain gauges are also embedded into the specimens to investigate the differences between the shrinkage in the core and on the surface of the test specimens. To show the importance of the time of the start of measurement and start of data evaluation, the recorded data are zeroed at the time of the start of measurement and at the time corresponding to the setting time of concrete. One of the goals of this paper is to show the correspondence between the measurements using two different measurement techniques and to examine the possibilities of their mutual substitution in the case of shrinkage measurement. The main reason for their mutual substitutability is to lower the costs of the measurements.

## 2. Materials

A special type of Self-Compacting HPC (SCC HPC) has been developed within the implementation of the project supported by the Ministry of Industry and Trade of the Czech Republic (project No. FV10588). This concrete has been primarily designed for the purpose of manufacturing the precast concrete members, primarily, with a high resistance to chloride and sulfate diffusion using locally available raw materials that reduce the costs of the concrete and final concrete members. The main demand on the composition of HPC, given by the producer of precast concrete members, was the usage of natural sand and coarse aggregate from the Zaječí quarry (Zaječí, Czech Republic). This type of spar-aggregate with smooth rounded grains is of quite low quality from the compressive strength point of view. One of the demands on the fresh-state concrete was a high flowability without changing within 20 min after mixing. Other demands were, for example, an average flexural tensile strength of 10 MPa without adding fibers and the compressive strength about 100 MPa and above. 

The composition of the concrete consists of CEM I 42.5 R Portland cement (Cement Plant Mokrá, Mokrá-Horákov, Czech Republic), Metakaolin in the amount of 15.6%, finely ground limestone (FGLS) in the amount of 20%, and multifunctional excellent admixture SST1 in the amount of 1.6% by cement mass. Admixture SST1 works as a water reducer, stabilizer, and workability keeping agent in one body. It was specially designed for a relevant HPC/UHPC set based on the same components. In principle, it is a blend of various polycarboxylates and polycarboxylate-ethers. Its detailed composition cannot be published because it is the producer’s know-how. Natural sand and gravel (mineralogical base-spar) with a maximum grain size of 8 mm is used as an aggregate. The aggregate-cement ratio (a/c) is 3.3 (by mass); the binder-aggregate ratio (b/a) is 0.65, the water-cement ratio (w/c) is 0.366 and the coefficient including all additives and admixtures is 0.281. The details of the composition of the SCC HPC are listed in Table 1.

The specimens intended for the shrinkage measurement and determination of the compressive and flexural tensile strength were manufactured for the purpose of the experiment. Details of the dimensions and curing conditions maintained during the experiment are stated in Section 3.

## 3. Experimental Investigation

The experimental analysis consisted of the following investigation steps: Determination of the shrinkage process of the test specimens cured under dry air (D-A) and autogenous conditions (AU).The comparison of the early-age shrinkage process recorded by inductive sensors (IS) leant against the movable head of the shrinkage molds (ShM) and vibrating wire-type strain gauges (W-SG) embedded inside the test specimens.The comparison of the overall shrinkage curves obtained by the continuous measurement using the wire-type strain gauges embedded inside the test specimens during the whole time of the experiment and the curves obtained from the measurement performed by the combination of the inductive sensor used for the early-age measurement and by the mechanical strain gauge (M-SG) used for the long-term measurement, placed (in predefined intervals) onto the markers embedded into the upper surface of the test specimens.Investigation of the influence of specimens’ size on the overall process of shrinkage. Two sets of specimens with dimensions of 100 mm × 60 mm × 1000 mm (ShM_100) and 50 mm × 50 mm × 300 mm (ShM_50) are manufactured for this purpose.Two different times *t*_0_ are considered for the data evaluation—in the first case, *t*_0_ is the time of the start of measurement, in the second case, *t*_0_ is the setting time determined based on the internal temperature curve.

### 3.1. Shrinkage and Mass Losses Measurement

The shrinkage process was monitored using the shrinkage molds with one movable head equipped with inductive sensors and wire-type strain gauges. The mass losses of the specimens molded in the ShM_100 molds were measured using a specially constructed weighing table which enabled the changes in mass to be measured continuously without handling the specimens during the early-age (see paragraph 3.1.3 for more details). The early-age mass loss of the ShM_50 specimens was determined as the mass loss of three test specimens placed together on the laboratory scale connected to the data logger. For the long-term measurement of the mass changes the laboratory scale with a sensitivity of 0.1 g was used for all specimens. 

During the early-age measurement (approx. 48 h), the inductive sensors and weighing table were connected to the QuantumX data logger made by Hottinger Baldwin Messtechnik GmbH (HBM, Darmstadt, Germany); the data were stored with a frequency of 5 Hz. The wire-type strain gauges were connected to the data logger made by DataTaker (Scoresby Vic, Australia) with the same storage frequency. Simultaneously, the internal temperature of the specimens was measured, the data were stored in the MS6 data logger (made by COMET SYSTEM, s.r.o., Rožnov pod Radhoštěm, Czech Republic) with a frequency of 1 min^−1^.

Two curing conditions were maintained during measurement—dry air and autogenous. The dry air conditions were represented by the laboratory conditions with an ambient temperature of 21 ± 1 °C and relative humidity of 55% ± 10%. To incite the autogenous conditions, the upper surface of the specimens placed in the ShM_100 molds was covered with a layer of Paraffin oil during early-age measurement. For the purpose of the long-term measurement, the specimens were wrapped with plastic foil (PE foil type) to avoid the humidity exchange between the specimens and environment.

In total, three sets of test specimens were manufactured for the purpose of the shrinkage measurement. Each set contained three test specimens. 

#### 3.1.1. Shrinkage Molds

Two types of shrinkage molds of different dimensions were used in the experiment—ShM_100 (produced by Shleibinger (Buchbach, Germany) [7] and ShM_50, designed and constructed at the Brno University of Technology (BUT), Faculty of Civil engineering (FCE), Institute of Building Testing, Brno, Czech Republic (see Figure 1, Figure 2a and Figure 3). Both molds are primarily intended for the early-age measurement which starts very early after placing the fresh-state concrete into the molds. The start of the measurement depends on the workability of the concrete. In this particular case, the measurement started approximately 1 h after placing the concrete into the molds. 

The measurement principle consists in the measurement of relative length changes along the longitudinal axis of the test specimen using the inductive sensor WA2 made by HBM with a measurement range of 2 mm. The sensor leans against the guiding bar of the movable head. The connection between the heads of shrinkage molds and concrete specimens is ensured by anchors mounted to both heads. The inside surfaces of the molds (excluding the heads) were covered with a PTFE foil, of 0.1 mm in thickness, to minimize the friction between the specimens and molds during the measurement. Inner sides of the heads were smeared with a Vaseline which was also used for sealing the interspace between the movable heads and the body of the molds. In this way, the length changes were measured for approximately 48 h. To enable the continuance in the long-term measurement, the molds were equipped with special markers which enabled the measurement of relative length changes after removing the specimens from the molds. 

In the case of the ShM_100 molds, the markers were embedded into the upper surface of the test specimens during the specimens’ manufacturing in such way that they formed the measurement base with a nominal length of 200 mm (see Figure 1). This base serves for the manual measurement of the length changes using the M-SG—Hollan—equipped with a digital indicator (see Figure 4b). The M-SG—Hollan apparatus is equipment designed and constructed by Assoc. Professor Karel Hollan at the BUT, FCE, Institute of Building Testing, Czech Republic. It is a type of transducer that uses the embedded markers placed in the specimens in the specific distance to evaluate the strain by using strain-gauges.

In the case of the ShM_50 molds, the markers were screwed into the anchors, which were embedded into the ends of specimens during their manufacturing, after the demolding of the specimens. The long-term measurement was performed using a dilatometer equipped with a digital indicator (see Figure 4c).

#### 3.1.2. Wire-Type Strain Gauge

Vibrating wire-type strain gauges made by GEOKON (Lebanon, NH, USA), model 4202, with an active gauge length of 51 mm and resolution of 0.4 µε, were placed in required locations before pouring the ShM_100 molds. The diameter of the circular end blocks is 15.5 mm. The W-SG sensors were embedded along the longitudinal axis of the specimens in the mid-length of the specimens (see Figure 1). The embedded strain gauges enable the length changes to be measured continuously during the whole time of specimens ageing. Refer to [8] for downloading the technical sheet of the sensors.

#### 3.1.3. Weighing Table

The weighing table is a specially constructed apparatus—a lever balance scale—developed at the BUT, FCE, Institute of Building testing, Czech Republic (see Figure 2b and Figure 3b). Currently, it is designed for the purpose of the simultaneous measurement of the changes in length and mass of the test specimens placed in the shrinkage molds ShM_100. This apparatus is intended especially for the measurement of the losses in the mass of the test specimens caused by the water evaporation during the early stage of the solidification process. It enables the mass losses to be monitored continuously without handling the test specimens. The apparatus consists of three separate weighing platforms joined with the lever mechanism equipped with the balancing weights and locking mechanism. The PW6C type single point load cells of an accuracy class C3, with a weighing capacity of 3 kg, placed under each platform are used for weighing (made by HBM). The lever mechanism of the weighing platforms enables a precise measurement of the changes in mass (caused only by the water evaporation) of the individual test specimen placed on the individual platform. The weighing platforms are placed on the rigid frame to minimize the influence of the ambient vibrations on the measurement process. The changes in the mass of the tests specimens are continuously recorded by the single point load cells connected to the QuantumX data logger. Refer to [9] for more details about the weighing table.

### 3.2. Mechanical Characteristics of Hardened Concrete

Three sets of test specimens cured under different conditions are manufactured for the purpose of determining the basic physical and mechanical characteristics of hardened SCC HPC, namely the bulk density (*D*), compressive strength (*f_c_*), and flexural tensile strength (*f_f_*). Each set contains six specimens with dimensions of 40 mm × 40 mm × 160 mm cast into the plastic polyethylene-type (Hakorit) molds (see Figure 5). For the purpose of the experiment, all test specimens were demolded after 24 h of ageing and stored under specific curing conditions. One set of specimens was cured in water, another set was PE-foil-wrapped and the last one was dry air cured until the testing time (including the first 24 h of ageing). These curing conditions were chosen intentionally to observe the influence of the curing conditions on the development of the strength parameters, which is useful for the interpretation of the shrinkage test results. The foil-wrapping and the dry air curing represent the curing conditions applied to the shrinkage test, namely the autogenous and dry air conditions. The water curing was taken as a reference curing method as given by many standards.

## 4. Results

The results of the performed experiment are displayed in tables and figures in this section. The basic characteristics of the hardened SCC HPC are summarized in Table 2 and Figure 6. The average values and sample standard deviations of bulk density (*D*), compressive strength (*f_c_*), and flexural tensile strength (*f_f_*) were calculated from three independent measurements for each testing set and concrete age. The results show that the values of strength parameters were affected by the curing conditions. In the case of compressive strength, the highest values were obtained for the specimens cured in water. The values determined at the age of 7 and 28 days were higher by about 8% and 2% compared to the values determined for foil-wrapped specimens, and by about 18% and 16% compared to the dry air cured specimens. The positive effect of the water curing on the values of flexural tensile strength was observed especially at the age of 28 days, when the values determined on the water cured specimens were by about 7% and 19% higher compared to the value obtained for the foil-wrapped and dry air cured specimens. However, the flexural tensile strength determined at the age of seven days was almost the same for all investigated sets and in this case, was not affected by the curing conditions, which is a positive finding with regard to the formation of the early cracks.

The results of shrinkage measurement are displayed in Figure 7, Figure 8, Figure 9, Figure 10, Figure 11 and Figure 12. The gray curves show the results of measurement performed on the specimens cured under autogenous conditions, while the black curves represent the results obtained from the measurements performed on the dry air cured specimens. The curves show the average values of investigated characteristics calculated from three independent measurements. The variability of the measurements is shown in Table 3. In all cases, the shrinkage and temperature curves are evaluated from the time when the measurement starts. The final setting time (ST) is determined based on the temperature curve as the time when the internal temperature achieves the half of the ascending part of the temperature curve. The procedure for the determination of the setting time is adopted from the procedure given by the ASTM standard [10].

Table 3 shows the average values (AV), sample standard deviations (SSD), and coefficient of variations (CoV) calculated for each measurement technique from three independent measurements performed simultaneously. Based on the variability of the results, the spread of the measured values and the difference between the values obtained by different measurement techniques is higher than the uncertainty of the measurements of measuring sensors used in particular measurement techniques. Therefore, the uncertainty of measuring sensors does not cause the spread in the measured results and does not change the discussion of results. The results show that the highest variability in the measured sets is at the very early ages of the test specimens and decreases with the concrete ageing. The variability of the measurements does not exceed 12% after reaching the setting time of concrete.

Figure 7 displays the process of early shrinkage and temperature development determined for the specimens with dimensions of 100 mm × 60 mm × 1000 mm. The results show that the absolute value of the internal temperature is affected by the curing conditions. The internal temperature measured for specimens cured under autogenous conditions shows steeper growth with the maximum value approximately 1.2 times higher than that for dry air cured specimens. The peak temperatures occur approximately at the same time. Concerning the shrinkage process, it can be stated that both sets of specimens started to shrink almost at the same time. Based on the results, it can be stated that in the case of the shrinkage process determined under autogenous conditions, the trend of the shrinkage process is not affected by the measurement technique. Similar findings are observed in the case of the shrinkage process determined under dry air conditions. Despite the fact that the IS sensor leant against the movable head of the shrinkage molds started to measure the shrinkage slightly earlier, the IS sensors and the embedded W-SG strain gauges give almost the same trend of the shrinkage curve for both autogenous and drying shrinkage. The shrinkage molds are able to start the shrinkage measurement very early due to their low-stiffness. Therefore, the changes in length caused by the drying of the upper surface of the test specimens can be recorded immediately after their molding. The specimens exposed to free drying show steeper growth of shrinkage due to the water evaporation.

The investigated SCC HPC show steep growth of the autogenous shrinkage with the value of approximately 25% of the total shrinkage determined for the dry air cured specimens at the age of 70 h (see Figure 7). Independently of the curing conditions, both sets of test specimens exhibit sudden braking of the shrinkage process followed by slight expansion at the time when the internal temperatures peak is reached. The subsequent decrease in the internal temperature causes a continuation of the shrinkage process; the steepness of the shrinkage curve is further influenced by the curing conditions (see Figure 7 and Figure 8).

Figure 8 shows the shrinkage curves recorded up to the age of 170 days. The results show a gradual increase in shrinkage values after demolding of both testing sets (to maintain the autogenous conditions, the test specimens are wrapped in the PE foil; the dry air cured specimens are left to dry freely from all surfaces). In the case of autogenous shrinkage, both employed measurement techniques (embedded W-SG and IS with continuation by the M-SG, see paragraph 3.1.1. and 3.1.2) give almost the same trend of the shrinkage curves up to the age of 60 days when the curve obtained by the M-SG measurement starts to deviate from the curve recorded by the embedded W-SG. In the case of dry air stored specimens, the different measurement techniques give slightly different results which are given by the different position of the gauging points—the drying occurs more rapidly on the surface which leads to a higher value of the shrinkage measured using the M-SG sensor. The differences in the shrinkage curves increase with the age of the specimen (see Figure 8).

The differences in the absolute values of the shrinkage recorded by different measurement techniques are summarized in Table 4. The relative values represent the average values of shrinkage calculated for the specific age of the concrete (see Table 3). Parameter “1” is considered for the measurement performed by IS+M-SG for both measurements under dry-air and autogenous conditions.

Figure 9 and Figure 10 show the shrinkage curves obtained for two different sizes of specimens. In both cases the shrinkage molds with an inductive sensor, for early age measurement, in combination with the mechanical strain gauge, for measurement after demolding, are used for recording the overall process of shrinkage during the concrete ageing. Figure 9 shows that the early age measurement is affected by the specimen’s size. The measurement performed on the smaller specimens with dimensions of 50 mm × 50 mm × 300 mm shows higher shrinkage with the value being 1.5 times higher than that recorded for the larger specimens with dimensions of 100 mm × 60 mm × 1000 mm at the age of 13 h (the time of reaching the peak temperatures). This difference in the shrinkage values persists up to the demolding time. After the demolding of the test specimens, the trend of shrinkage shows that the specimens of both sizes continue approaching the same ultimate shrinkage value (see Figure 10). The values of shrinkage determined for individual testing sets differ by about 10% at the age of 170 days.

The differences in the absolute values of the shrinkage recorded for different sizes of specimens are summarized in Table 5. The relative values represent the average values of shrinkage calculated for the specific age of the concrete (see Table 3). Parameter “1” is considered for the measurement performed by IS+M-SG_ShM-100.

Figure 11 shows the process of mass losses determined for all testing sets. The specimens cured under dry air conditions show almost the same trend of free desiccation independently on the size of the test specimens. The absolute value of loss in mass is about 2% at the age of 170 days. The decrease in mass is also recorded for the specimens wrapped in the PE foil. The absolute value of the mass loss is, in this case, about 0.2%.

Figure 12 shows the relationship between the shrinkage and mass losses process determined for the specimens of different dimensions cured under dry air conditions. The steep increase in the shrinkage accompanied by low water evaporation is recorded at the initial part of the curve. The sudden breaking of the curve corresponds to the time of reaching the peak temperatures after which the gradual increase in shrinkage values and decrease in mass of the test specimens occur.

## 5. Discussion

There are two main aspects which are discussed in this section. 

The first aspect deals with the measurement techniques used for the experiment and with the time when the data evaluation of the measurement started (*t*_0_).

In this paper, the authors used molds with a movable head equipped with an inductive sensor and markers for embedding into concrete, and they used a vibrating wire-type strain gauge with a low stiffness for the measurement of both the autogenous and drying shrinkage. 

Concerning the autogenous shrinkage, the experiment shows that both testing devices give the same trend of the shrinkage process under the chosen test configuration up to the age of approximately 60 days (see Figure 7 and Figure 8). Despite the fact that the shrinkage under autogenous conditions reached very low absolute values, the variability of the measurement of both measurement techniques did not exceed 20% for the measurements recorded before the setting time and decreases with the concrete ageing. The differences in the average values obtained by the different measurement techniques were about 20% in the time interval from 12 h to 60 days (see Table 3 and Table 4). After the age of 60 days, the curve measured by the mechanical strain gauge (placed onto the markers embedded into the top surface of the specimens) started to deviate slightly from the curve recorded by the embedded W-SG. When *t*_0_ was considered as the time of the start of shrinkage measurement, the value of autogenous shrinkage recorded during the experiment was approximately 470 µm/m measured by the embedded strain gauge and 670 µm/m measured by the combination of IS and M-SG, both were measured at the age of 170 days. 

In the case of drying shrinkage, the shrinkage curves obtained by the embedded strain gauge exhibits shrinkage with the steady-state value of 1160 µm/m was reached at the age of 70 days. The drying shrinkage measured on the top surface with the value of about 1500 µm/m at the age of 170 days was not yet stabilized. The highest difference in the average values obtained by different measurement techniques was recorded at the very early ages up to the time of setting when the difference between the average values was about 30%. In the time interval from 12 h to 60 days, the average values differ in the range of 15–20%. After 60 days, the difference in shrinkage values gradually increased (see Table 3 and Table 4). 

Based on the results it can be stated, that both measurement techniques enable the shrinkage strain to be recorded during the whole time of the solidification process, including the plastic and semi-plastic stage. This is advantageous in terms of determination of the autogenous shrinkage and shrinkage induced cracking. The main difference between these two measurement techniques is the position of the sensors, which influences the absolute value, especially in the case of the drying shrinkage because of the different rate of desiccation of the core and surfaces of the specimens. It can be expected that in the case of drying shrinkage, the difference in the shrinkage values grows with the distance from the core to the surface of the test specimens. However, this statement needs to be verified by further testing. Another difference between these techniques is in the principle of the measurement and related stiffness of the measurement devices. These characteristics influence especially the time when the measurement is started. In this particular case, the shrinkage molds start to measure earlier than the embedded wire-type strain gauge because the sensor is leaned against the movable head which enables recording of the shrinkage strain caused by early desiccation of the upper surface of the specimens. 

If the experimental results of the shrinkage measurement are compared to each other, all measurements have to be zeroed at the same time. In this paper, the data are primarily zeroed at the time when the measurement is started. It means that the recorded shrinkage curves include the shrinkage which takes place in the plastic and semi-plastic stage of solidification process (see Section 4). Most of the researchers start the measurement evaluation at the initial or final setting time, sometimes even at the time of reaching the maximum of the internal temperature [11] which may lead to incorrect evaluation of the real value of the shrinkage, as reported in [5] and [12]. 

If the data obtained by the experiment described herein are evaluated from the final setting time and the time when the temperature reaches its peak, the absolute value of the autogenous shrinkage is lower by about 60 µm/m and 100 µm/m respectively in the case of data recorded by the embedded W-SG. In the case of data recorded by a combination of IS and M-SG, it is lower by about 90 µm/m and 140 µm/m respectively. The difference in the absolute value of the autogenous shrinkage due to the different initial time approaches of evaluation is between 12–21%. The evaluation of the absolute value of drying shrinkage is more affected by the time of the start of the evaluation. Concerning the data recorded by the embedded W-SG, the absolute value of the drying shrinkage is lower by about 460 and 600 µm/m when the time *t*_0_ is the final setting time and the time when the temperature reaches its peak, respectively. Concerning the measurement performed with the combination of IS and M-SG, the values are lower by about 660 and 760 µm/m. In this case, the reduction of the absolute value of the drying shrinkage due to the different approach of evaluation is between 40–50%. For the illustration, the curves obtained by measurement using the combination of IS and M-SG evaluated at different time *t*_0_ are shown in Figure 13. The dashed lines represent the shrinkage curves obtained from data zeroed at the start of measurement.

The second aspect discussed herein deals with the size of specimens and related influence of different time (*t*_0_) on the final conclusion about the size effect. The final shrinkage curves obtained from the evaluation of the data at the time of the start of measurement and at the setting time are shown in Figure 14, for different size specimens, cured in dry-air conditions and measured by the combination of IS and M-SG. The dashed lines represent the shrinkage curves obtained from data zeroed at the start of measurement. The yellow and gray lines represent the shrinkage curves for smaller size specimens zeroed at the time corresponding to the minimum internal temperature (T_in, min_) and to the setting time respectively. Similarly, the red and black lines represent the shrinkage curves for larger size specimens zeroed at the time corresponding to the T_in, min_ and to the setting time respectively.

Based on the results of performed experiment, it can be stated that the effect of the specimen size is strongly affected by the time when measurement and data evaluation starts. The results presented in Figure 9, Figure 10 and Figure 14 show more rapid shrinkage recorded for the specimens with dimensions of 50 mm × 50 mm × 300 mm immediately after the start of measurement. This effect is related to the internal forces induced by a capillary suction in the capillary pores [13] which draw the mass of concrete. Because the specimens 50 mm × 50 mm × 300 mm were 8 times smaller in volume compared to the volume of specimens 100 mm × 60 mm × 1000 mm, the internal driving forces were able to draw the mass of concrete earlier which influenced the initial part of the measured shrinkage curve. If the data recorded before the setting time are neglected and are not taken into account for the evaluation, different conclusion for the size effect are obtained (see Figure 14). The measured curves show quite a small effect of the size of specimens on the shrinkage process up to the age of about 60 days when the curves evaluated for both sizes of specimens start to deviate slowly one from another. These results also show higher shrinkage for the bigger specimens. A similar effect with higher absolute values of shrinkage was observed when the data were zeroed at the time corresponding to the minimum internal temperature of concretes (see Figure 14). This is in contradiction with the results shown in Figure 10 and Figure 14 for the shrinkage curves which are evaluated from the start of measurement. Figure 15 shows curves of the relative values of shrinkage (a ratio of shrinkage of smaller specimens to shrinkage of larger specimens) during the concrete ageing determined for the shrinkage curves zeroed at the time of the start of measurement and at the time corresponding to the minimum internal temperature. The results show that the process of shrinkage is strongly affected in both cases by the specimen’s size up to the time of reaching the temperature peak. After that, the relative values of the shrinkage are stable up to the demolding time (see Figure 15). During the time interval from demolding to the age of 50 days, the relative values of shrinkage of different size specimens are about 1.0 for the shrinkage curves zeroed at the time of T_in, min_. The gradual decrease of the relative values from 1.0 to 0.9 is observed within the time interval from 50 days to 170 days. For the shrinkage curve zeroed at the time of the start of measurement, there is a subsequent decrease in the ratio from 1.4 to 1.1 in the time interval from demolding up to the age of concrete of 170 days.

The size effect for the shrinkage measurement is closely associated with the effective thickness of the specimens characterized by the volume-surface ratio, defined as *2V*/*S*, where *V* is the volume of the specimens and *S* is the specimens’ surface exposed to drying. Its relation to shrinkage is based on the diffusion analysis of drying [14]. The results reported by M. Samouh et al. [14,15] show no significant effect of the size of the specimen on the shrinkage process if the specimens with similar effective thickness are used. These results are supported by H. Ba et al. [16] which deals with the specimens of high differences in the effective thickness and show a high difference in the shrinkage process in dependence on the sizes of specimens. Note, that both researchers started their measurement after demolding of the test specimens and the very early shrinkage is not reflected in their results. 

For the purpose of the evaluation of the results, the effective thickness of the specimens of 100 mm × 60 mm × 1000 mm and 50 mm × 50 mm × 300 mm, respectively, was calculated for the measurements performed before and after demolding of the test specimens. The effective thickness was 120 and 100 for larger and smaller specimens respectively in the case of measurement performed before demolding (only the upper surface was exposed to free desiccation) and 36.1 and 23.1 in the case of measurement performed after demolding (all surfaces were exposed to free desiccation). The authors used the ratios of the effective thickness as the indicator of prediction of the size effect in the case of the shrinkage process. The authors expect that the effective thickness ratios and shrinkage ratios are similar at these particular stages of specimens ageing. The ratio of the effective thickness of the smaller and larger specimens used for the experiment presented herein is 0.83 and 0.64 in the case of measurement before and after demolding of the specimens respectively. This implies that the small effect of the specimens’ size is expected during the measurements before demolding and higher size effect is expected after demolding of the test specimens. However, the results presented in Figure 15 show that different effect of the specimens’ size was recorded for different times *t*_0_. Because of the early start of measurement, the shrinkage curves include the plastic and semi-plastic type of shrinkage which is not fully proportional to the rate of drying (see Figure 12). In this case, it is crucial to relate the conclusions to the time of the start of measurement and data evaluation. The discussion above shows that the problem of the size effect has to be investigated in more details in the case of the measurements of the shrinkage which cover the plastic and semi-plastic stage of solidification process. 

Finally, the analysis of cost made by the authors shown that the cost of measurement might vary depending on the commercial conditions in each country. The embedded wire-type strain gauges are single-use units and their cost depends on the parameters of the particular model. On the contrary, the shrinkage molds are re-usable and the cost of the measurement can be reduced by more than 50% compared to the measurement using the wire-type strain gauges. The main disadvantage of the shrinkage molds is the initial cost of the molds and inductive sensors. In this particular experiment, the overall cost of one measurement set containing three molds equipped with the inductive sensors was about 12 times higher than the testing set containing three test specimens with embedded strain gauges. However, considering the simpler setting of the gauges, the associated measuring equipment and a large number of tests associated to batch-test procedures in testing activities, the use of shrinkage molds results in a highly competitive method in the pursue for cost-effective HPC techniques.

The next step of the investigation program is to process the data for the numerical analysis and to compare the experimental results with the shrinkage curves predicted using available numerical models [17]. The initial partial results indicate that the B4 model designed by the research group of Professor Z. P. Bazant [18] should be the most suitable for numerical simulation. This model offers both the simplified and the advanced analysis which enables to reflect the real composition of the concrete. Unfortunately, some of the components e.g., the spar-aggregate, Metakaolin, and specific type of admixture are not included in the actual version of the B4 model. In addition, plastic shrinkage is not fully implemented in the actual version. In this case, the authors plan to compare the current results of experimental analysis and numerical prediction using the simplified analysis implemented in B4 model. The authors expect that this step will lead to the specification of the correction factors which could improve the correspondence of the experimental and numerical analysis. 

Another step of the investigation is to extend the experiment with the measurements focused on the cracking induced by the plastic shrinkage of the SCC HPC.

## 6. Conclusions

The paper presents the results of the experimental analysis focused on the determination of the shrinkage process using two different types of measurement techniques—sensors embedded in the core and into the upper surface of the specimens. Two different times *t*_0_ were chosen for the start of evaluation of the recorded data to show the significance of this step of the experimental analysis, and to compare with other literature results which disregard this point. In addition, the size effect in relation to the time *t*_0_ is discussed in the paper.

Based on the results, the following main conclusions can be drawn:Both measurement techniques used for the experiment are suitable for measurement of the overall process of shrinkage—since the placing of the fresh-state concrete up to the long-term measurement of the length changes of the hardened concrete (including the autogenous shrinkage).Both measurement techniques gave the same trend of the shrinkage curve up to the age of ca 60 days. In this particular case, the difference in the average values was about 20% in the case of the autogenous shrinkage in the time interval from 12 h to 60 days. The average values measured around and before the time of setting differed by 30% and more. The difference in the average values of drying shrinkage due to the different measurement techniques was about 30% around the time when the internal temperature started to rise and decreased with the age of concrete up to the 60 days when the difference between the measurement techniques was about 10%. After this time the differences increased due to the different positions of the gauging points. This finding is important in relation to long-term measurements because typically, the shrinkage measurement is terminated in the time when the shrinkage values stabilize. From the long-term measurement point of view, the embedded sensors could give an underestimated or incorrect value of shrinkage for the concrete structures with surfaces exposed to drying because of the different rate of desiccation of the core and surfaces of the specimens.The performed experiment shown that the wire-type strain gauge was able to start the measurement of the shrinkage around the time of setting of the concrete (at the age of concrete more than 4 h). On the contrary, the shrinkage molds were able to start the measurement of the shrinkage around the time when the internal temperature of the concrete starts to rise (earlier than 4 h). Concerning the cost of measurement, it may vary depending on the commercial conditions in each country. In this particular experiment, the overall cost of one measurement set containing three molds equipped with the inductive sensors was about 12 times higher than the testing set containing three test specimens with embedded strain gauges. However, considering the simpler setting of the gauges, the associated measuring equipment and a large number of tests associated to batch-test procedures in testing activities, the use of shrinkage molds results in a highly competitive method in the pursue for cost-effective HPC techniques.It was proven that the time *t*_0_ (time of the start of data evaluation) has a fundamental influence on the final values of the shrinkage of investigated SCC HPC. The difference in the absolute value of the shrinkage due to the different initial time approaches of evaluation can be up to 50% in the case of the drying shrinkage measurement and up to 20% in the case of autogenous shrinkage measurement.It was observed that the time *t*_0_ has a significant impact on the conclusions concerning the size effect. Especially, the measurement during the plastic stage is influenced by the specimens’ size. If the measurement starts as early as possible after the placing of the fresh-state concrete into the molds, the steepness of the shrinkage curve and also the final value of the total shrinkage is strongly affected by the specimens’ size due to the capillary suction. The results presented in Figure 15 show that different effect of the specimens’ size is recorded for different times *t*_0_. Because of the early start of measurement, the shrinkage curves include the plastic and semi-plastic type of shrinkage which is not fully proportional to the rate of drying (see Figure 12). In this case, it is crucial to relate the conclusions to the time of the start of measurement and data evaluation.

## Figures and Tables

**Figure 1 materials-12-02730-f001:**
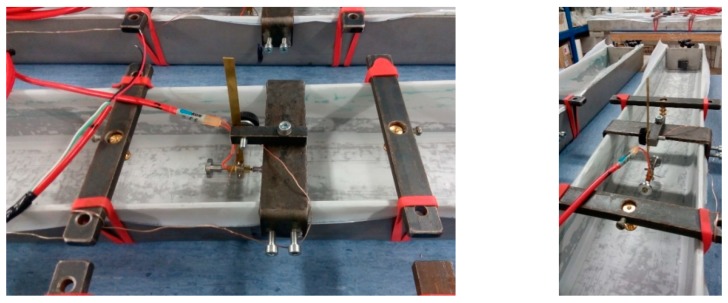
Shrinkage molds 100 mm × 60 mm × 1000 mm before concreting: Position of the wire-type strain gauges and the markers intended for measurement using the mechanical strain gauge.

**Figure 2 materials-12-02730-f002:**
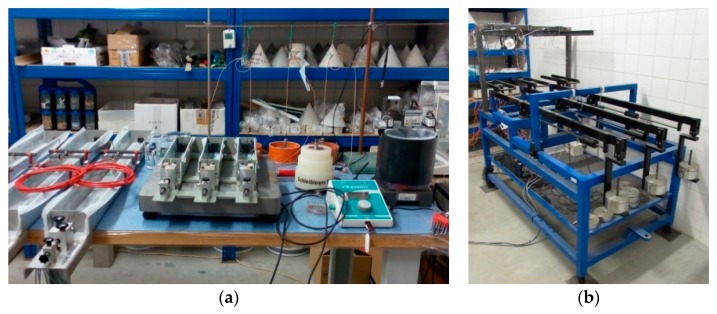
Overall view before concreting (**a**); weighing table for ShM_100 mm × 60 mm × 1000 mm (**b**).

**Figure 3 materials-12-02730-f003:**
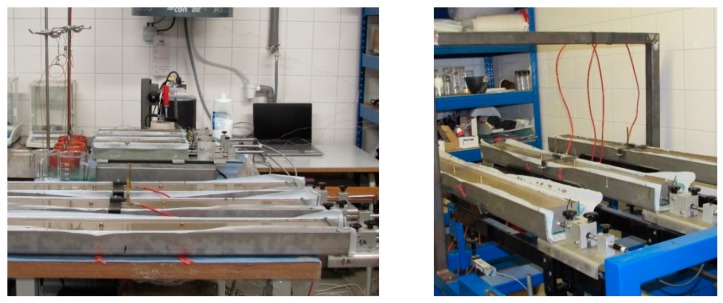
Overall view before start of measurement.

**Figure 4 materials-12-02730-f004:**
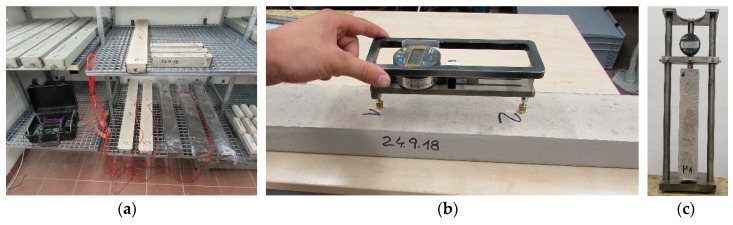
Illustrative photo of long-term measurement: Overall view (**a**); M-SG (Hollan) for specimens 100 mm × 60 mm × 1000 mm (**b**); dilatometer for specimens 50 mm × 50 mm × 300 mm (**c**).

**Figure 5 materials-12-02730-f005:**
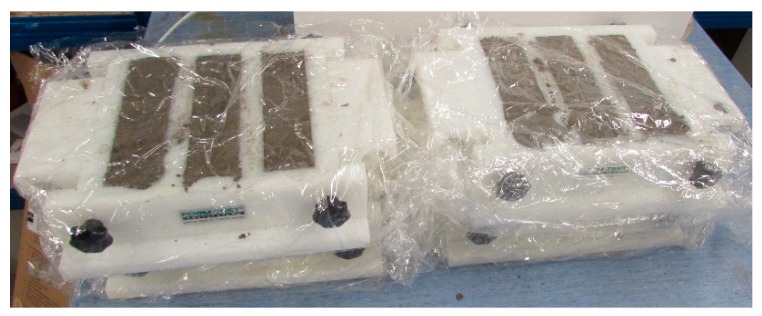
Test specimens in Hakorit molds (PE-foil wrapping is used).

**Figure 6 materials-12-02730-f006:**
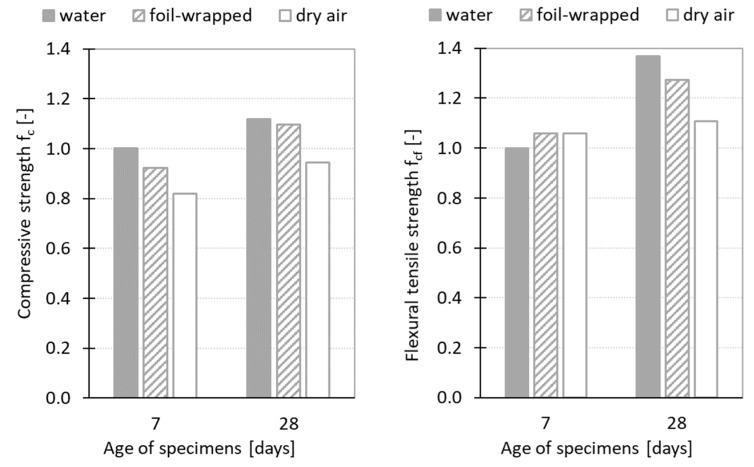
Characteristics of hardened Self-Compacting High-Performance Concrete: Relative values of compressive strength and tensile strength: 1 = water cured specimens at the age of 7 days.

**Figure 7 materials-12-02730-f007:**
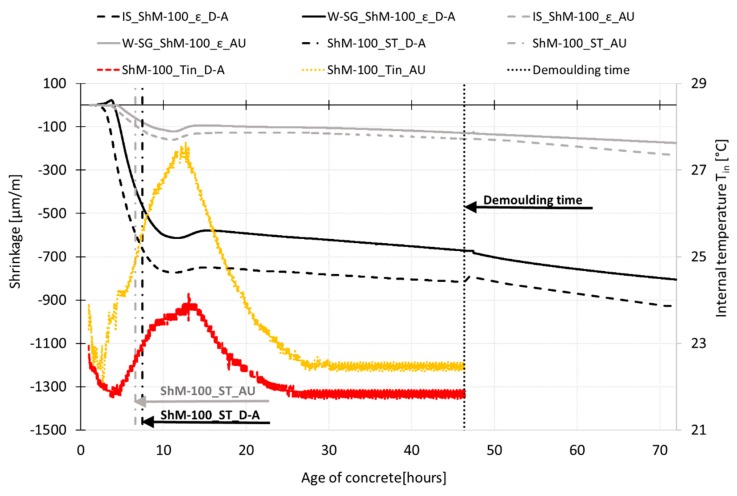
Process of early-age shrinkage recorded by inductive sensor of shrinkage molds (IS_ShM) and embedded strain gauge (W-SG_ShM) of specimens cured under dry air (black lines) and autogenous (grey lines) conditions. The recorded internal temperature (T_in_) of specimens (color lines) is also displayed.

**Figure 8 materials-12-02730-f008:**
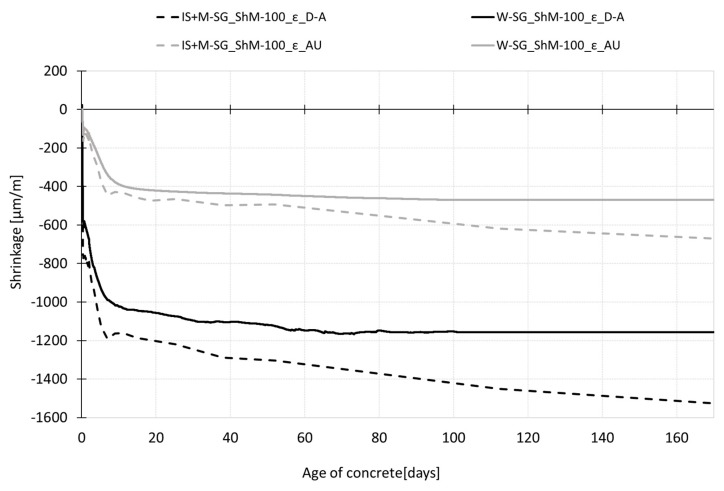
The overall process of shrinkage (ε) recorded during the whole time of measurement for specimens cured under dry air and autogenous conditions.

**Figure 9 materials-12-02730-f009:**
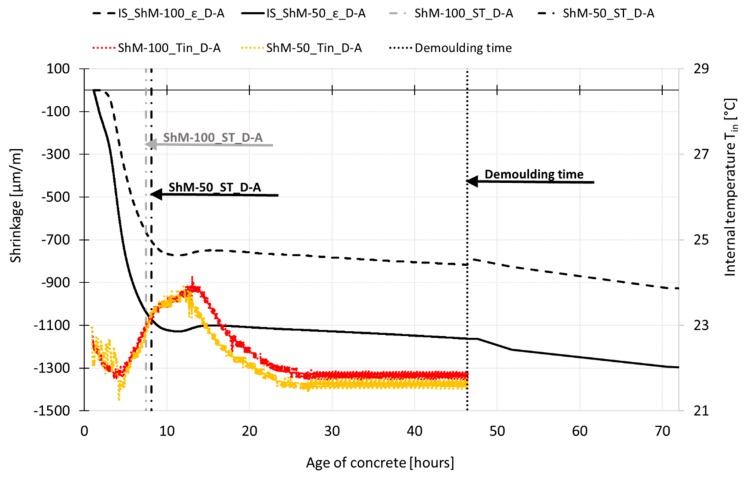
Process of early-age shrinkage recorded by inductive sensor of shrinkage molds (IS_ShM) of different dimensions and process of internal temperature (T_in_) of specimens cured under dry air conditions.

**Figure 10 materials-12-02730-f010:**
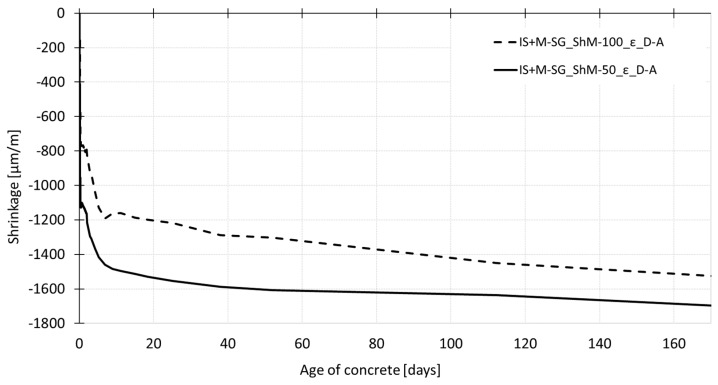
The overall process of shrinkage (ε) during the whole time of measurement for specimens of different dimensions cured under dry air conditions.

**Figure 11 materials-12-02730-f011:**
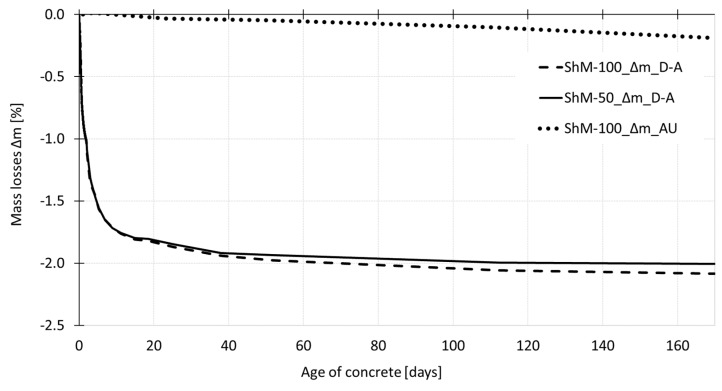
The overall process of mass losses (Δm) determined for specimens of different dimensions cured under dry air conditions.

**Figure 12 materials-12-02730-f012:**
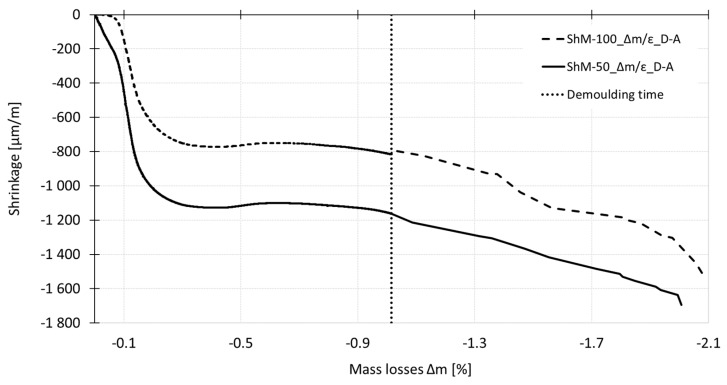
Relationship between shrinkage (ε) and mass losses (Δm) determined for specimens of different dimensions cured under dry air conditions.

**Figure 13 materials-12-02730-f013:**
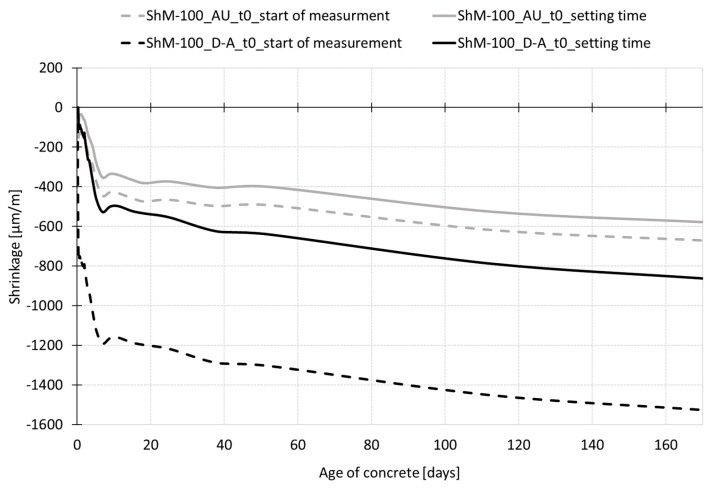
Shrinkage curves for the specimens 100 mm × 60 mm × 1000 mm: zeroed at different time *t*_0_.

**Figure 14 materials-12-02730-f014:**
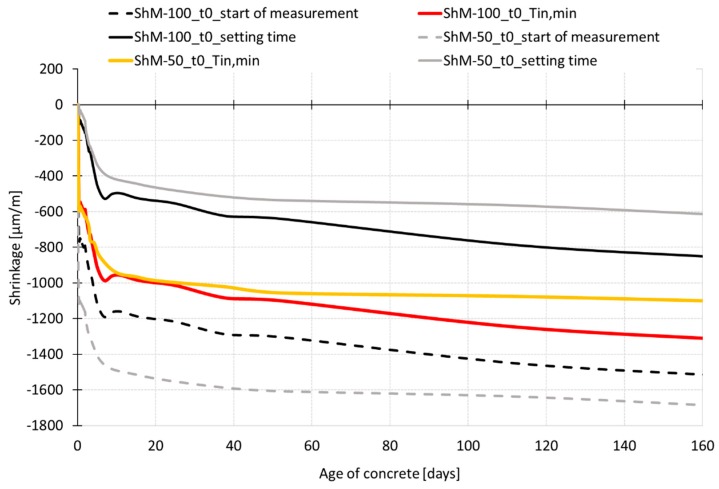
Effect of specimens’ size: zeroed at different time *t*_0_.

**Figure 15 materials-12-02730-f015:**
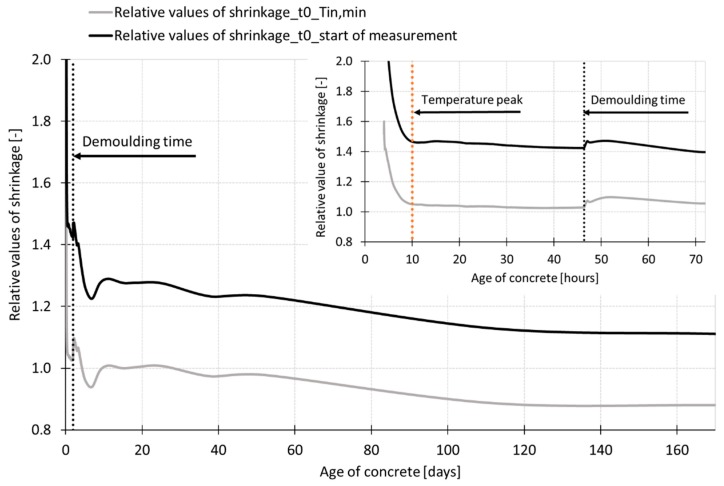
Effect of specimens’ size: Relative values of shrinkage at different time *t*_0_.

**Table 1 materials-12-02730-t001:** Composition of fresh High-Performance Concrete: Relative amount per mass.

Components	Relative Amount
CEM I 42.5 R (Mokrá, CZ)	1.000
Metakaolin “Mefisto K05” (CLUZ a.s. CZ)	0.156
Finely ground limestone V	0.200
Total amount of water	0.366
Multifunctional excellent admixture SST1	0.016
Sand 0/4 mm (Zaječí, CZ)	1.898
Aggregate 4/8 mm (Zaječí, CZ)	1.397

**Table 2 materials-12-02730-t002:** Bulk density (*D*), compressive strength (*f_c_*) and flexural tensile strength (*f_f_*) of the hardened Self-Compacting High-Performance Concrete: Average value (sample standard deviation).

	*D* [kg/m^3^]		*f_c_* [MPa]		*f_f_* [MPa]	
Curing Conditions	7 Days	28 Days	7 Days	28 Days	7 Days	28 Days
Water	2390 (6.9)	2390 (7.8)	95.7 (1.82)	107.1 (2.91)	8.4 (0.19)	11.5 (0.34)
Foil-wrapped	2370 (14)	2370 (13)	88.2 (0.45)	104.9 (4.24)	8.9 (0.7)	10.7 (0.86)
Dry air	2360 (15)	2350 (24)	78.4 (2.28)	90.4 (2.52)	8.9 (0.27)	9.3 (0.79)

**Table 3 materials-12-02730-t003:** Shrinkage values of the SCC HPC: Average value (AV), sample standard deviation (SSD), coefficient of variation (CoV).

		IS+M-SG_ShM-100		W-SG_ShM-100_D-A		IS+M-SG_ShM-50
		D-A	AU	D-A	AU	D-A
4 h	AV [µm/m]	−203	−12	n/a	n/a	−530
	SSD [µm/m]	71	1			112
	CoV [%]	35	9			21
Setting time	AV [µm/m]	−664	−90	−465	−58	−1072
	SSD [µm/m]	60	16	30	8	130
	CoV [%]	9	18	6	13	12
12 h	AV [µm/m]	−771	−149	−613	−115	−1126
	SSD [µm/m]	59	15	10	10	131
	CoV [%]	8	10	2	8	12
Demolding time	AV [µm/m]	−817	−154	−682	−132	−1163
	SSD [µm/m]	64	17	9	11	128
	CoV [%]	8	11	1	8	11
70 h	AV [µm/m]	−926	−227	−798	−171	−1295
	SSD [µm/m]	58	20	8	12	105
	CoV [%]	6	9	1	7	8
60 days	AV [µm/m]	−1323	−498	−1147	−449	−1611
	SSD [µm/m]	13	11	1	13	126
	CoV [%]	1	2	0.1	3	8
170 days	AV [µm/m]	−1526	−662	−1157	−470	−1696
	SSD [µm/m]	13	15	3	13	134
	CoV [%]	1	2	0.3	3	8

Note: n/a—the wire-type strain gauges are not able to start the measurement at the specimens’ age of 4 h due to their stiffness.

**Table 4 materials-12-02730-t004:** Shrinkage values of the SCC HPC: Relative average values of shrinkage determined by different measurement techniques.

	IS+M-SG_ShM-100		W-SG_ShM-100	
	D-A	AU	D-A	AU
12 h	1	1	0.79	0.77
Demolding time	1	1	0.84	0.85
60 days	1	1	0.87	0.90
170 days	1	1	0.76	0.71

**Table 5 materials-12-02730-t005:** Shrinkage values of the SCC HPC: Relative average values of shrinkage determined for different size of the specimens.

	IS+M-SG_ShM-100	IS+M-SG_ShM-50
	D-A	D-A
12 h	1	1.46
Demolding time	1	1.42
60 days	1	1.22
170 days	1	1.11

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
