# Peer review of "Cost-Effective High-Performance Concrete: Experimental Analysis on Shrinkage"

_materials, 2019, doi:10.3390/ma12172730_

Round 1

Reviewer 1 Report

The paper on Cost-Effective High-Performance Concrete: Experimental Analysis on Shrinkage, by B. Kucharczyková, D. Kocáb , P. DanÄ›k and  I. Terzijski, shows the results of a systematic program to measure the shrinkage of one specific HPC material.

Despite the necessity to probe that a certain mixing for HPC is adequate for construction purposes, it exists the underneath discussion of correctly  characterizing the HPC shrinkage for both, the effective comparisons with other HPCs, and the overall understanding of the solidification process, specifically the initial plastic and semi-plastic stage.
The paper contains valuable information and detailed description of  the measuring techniques applied. The reviewer has a positive recommendation for the publishing of the paper, although authors must make an effort to clear up important points not addressed in the paper, and to solve some discussion in a sound manner.  

The language used and the explanations of authors are carefully written. Some (too) technical details can be better addressed to help the reader to follow the procedures.   

The requests do not demand extra content, but adding missing information or clearing up some important parts. Surely authors can solve the different points effectively.

After a carefully consideration by the authors about the following points, the paper can be a valuable piece of information worth to be published in Materials.
(POINT A)
 Authors present a collection of measurements with specific techniques. The techniques applied, measuring at the core and the near surface are quite interesting and authors show an efficient know-how for tackling the problem. The results, indeed show ranges of agreement and also differences.  However, authors do not discuss the purpose of using  the selected techniques: comparison with other authors?  discussion of techniques and results ? comparison of methods to demonstrate compatibility VS costs ?. At some point authors should state what they learnt about using the two techniques, and what is the recommendation for future measurements.
(POINT B)
Authors do not quote uncertainties of the measurements, nor the spread in the measured results. It is never cleared up (to the reviewer limit) how many specimens were measured to stand a shrinkage value. And if only one specimen was measured, the uncertainty is unavoidably of interest. Indeed, authors quote similarity of results without referring to any quantitative value, or relation with the underneath uncertainty.  

Considering the experimental nature of the core of the paper, this information is critical, and without that the paper should be rejected. In any case, the demand in not too tough: clear up uncertainties, even to a conservative limit (10%?); quote numerical differences when comparing two curves, and comment on the implication that the number of specimens may have on the discussion.

(POINT C)

The information appearing  in the Figure 7 for the first time, is really hard to digest. Despite the option of authors to use grey-scale for the plots, which is correct, for the case of showing temperature and strain, please consider to use color (for instance, for temperature curves). That will save a lot of time and effort to any reader, helping to focus into the information and not the puzzle of too similar lines...

(POINT D)
To the reviewer point of view, the weakest point affects to the main discussion and conclusion of the paper. Authors discuss in lines L396-495 the comparison of the new results relating effective ratios to other literature values. However, the discussion is really tough to follow,  and requires that authors make an effort to make an effective and sound discussion to conclude with their proposal.

Authors should address the above points (A-B-C-D) through the paper. More comments follow below.

COMMENTS 

In the introduction, about lines  L45-ff, even if authors do not expand to more references, at least mention something the kind of  ' [2] and references therein'.  Then, readers can look for updated information at the right spot. 

L55
... see chapter 2
what is this as reference ?  authors mention lines above references 2 and 3, and concepts like excellent  admixtures and then, chapter 2.
please make a consistent discussion and do not expect that readers have to guess how to obtain the straight information which authors want to use.

L68-69
...the low stiffness type must be used; the length must be appropriate

>> the low stiffness GAUGE type must be used; ITS length must be appropriate

 L77
...It is well known that different measurement techniques give different shrinkage curves.

At this point, maybe authors can connect with the measurement techniques applied, or the time zero issue. In any case, authors must  Also consider that there are no references at this point where the reader can see the differences and the techniques etc.  If the aforementioned references are adequate, please, mention explicitly.    

TABLE  1

in the previous discussion the relative amounts of components have been presented with very different references. Table-1 is the chance to set a unique reference.   Table caption mentions 'per m3' , and table entry is for 'relative amount'.  please state if the numbers are per mass or volume.         In the text there is a list of commercial brands. Please use an uniform format at the text, kind of  QuantumX (Hottinger Baldwin Messtechnik GmbH, Germany)   L147  QuantumX : HBM ?

L148  DataTaker : Thermo Fisher ?

L150 Comet data logger

L165 inductive sensor WA2 : KEYENCE ?

L187 GEOKON

L201 PW6C type single point load cells  HBM ?

L218 y Fig-5  Hakorit plastic : commercial name of a generic plastic ?!
 L155
... wrapped in the PE foil to ...

consider to be a bit more specific: ... wrapped with plastic foil (PE foil type) 
 L160 and L195
BUT, FCE, Institute of Building testing (COUNTRY)
quote the full names under the acronym and country the first time is used. 
 L178
M-SG – Hollan
please define what is this system. if Hollan is commercial brand, apply the aforementioned format. 
L178
fig b    check the label :  fig 2-b ??
L187
... active gauge length of 51
please quote UNIT (length ?) 
L194
The weighing table is specially constructed apparatus
>>
The weighing table is A specially constructed apparatus
Section 3.1.3
maybe at this section, authors should clear up about uncertainties  of data, number of specimens.
 L220
...another set is foil wrapped ...
>> ...another set is PE-foil wrapped ...
FIGURE-5
caption: mention that PE-foil wrapping is used
L230
...displayed in tables and figures below
...displayed in tables and figures IN THIS SECTION
 L231
consider to mention what magnitudes are discussed, and relate to Table and symbols. Also mention that the data was measured at two different time moments (7 and 28 days), and that the mean values and deviations were defined from the specimens of each set, being tested 3 at each reference time (this si supposed by the reviewer..., please clarify this point)
 TABLE-2
consider to include a line or blank space between sets for better visualization.
the same, but less necessary, in between magnitudes.
caption: indicate the magnitudes measured, and relate the used symbols D  Fc   Ff
FIGURE 7
refer to point D
FIGURE 7 caption
... specimens cured under dry air (black lines) and autogenous conditions (grey lines).
L272
at the age of 70 days (see Fig. 7).
>>
CHECK: 
at the age of 70 HOURS (see Fig. 8) ???   otherwise the discussion makes no sense....
 L275
typo: CAUSES 
L289
... give almost the same shrinkage curves up to the age of 60 days...  
authors do not quantify the 'almost', nor compare with uncertainties of measurement. 
at least should set a limit (10% ??) as reference to speak of  'similar' and 'different'.  
refer to POINT B
L332
...both testing devices give very similar results...
within 5% 10% 15% (to be compared in relation with measurement uncertainties).
refer to POINT B
 L340
...exhibits lower shrinkage...
it is higher shrinkage, but measured as negative value, the reviewer supposes. maybe re-phrased as:
... exhibits higher shrinkage (lower absolute values) ...
L332-345
the discussion is confusing.
both DA and AU methods show the same behaviour: the embedded W-SG measurements level off in long-term, but the surface M-S shows an important time slope.
authors have the chance to compare if the two techniques are compatible to provide the same information, or why the differences are of importance.
 L350
... as reported in [5] AND [12].
 L353
... is lower by about 60 μm/m and 100 μm/m in...
>>
... is lower by about 60 μm/m and 100 μm/m RESPECTIVELY in...
L355
... it is lower by about 90 μm/m and 140 μm/m.
>>
... it is lower by about 90 μm/m and 140 μm/m RESPECTIVELY.
L356
it is necessary to help the reader to keep the focus of the discussion.
... due to the different approach...
>>
... due to the different INITIAL TIME approachES ...
L372
it is necessary to help the reader to keep the focus of the discussion.
... are shown in Fig. 14.
>>
... are shown in Fig. 14, for different size specimens cured in dry-air conditions, and measured by the combination of IS and M-SG.
This way figs 13 and 14 are directly comparable.  
  L385
... The resulted curves show quite a small effect ...
>> ... The MEASURED curves show quite a small effect ...
and again quote what 'small' is : 10% ? 15% ?
refer to POINT B
 L391
...specimens with similar effective thickness are used.
authors refer to a new concept that will be used again later.
please define at this point, even briefly, to avoid the necessity of reading the original reference for only this concept.
L396-495
the discussion is really tough to follow, with ratios which are not explained and values which are or not significant. therefore the reader can hardly agree with the main result of the paper: (L405) it can be stated that the shrinkage which takes place during the plastic stage of the solidification process affects the results of the size effect significantly.
what is the relation between the ratio of the shrinkage and the results presented? why the numbers obtained drive to any conclusion about the   effective effect or not of the size ?
authors should invest few extra lines to make a sound and consistent discussion that, eventually, drives to the presented conclusion.
L418
...This model offers beside the simplified also the advanced analysis which enables ...
consider to re-phrase:
This model offers both the simplified and the advanced analysis, which enables...
L420-ff
Authors propose a model (B4) but also mention its limitations:  
... Unfortunately, some of the components...  In addition, the plastic shrinkage is not fully implemented ...
Authors should add a line at this point to mention if they plan to update the model, or if they expect that despite the limitations they can obtain a consistent discussion when compared to the measured results.  Otherwise, can the state-of-the-art models explain the effects found in the measurements ?
L427 VS L435
the techniques should be distinguished at the beginning, the first time they are mentioned (L427), not later (435):    
(embedded in the core or into the upper surface of the specimens).
L428
...to show the significance of this step of the experimental analysis.
please set clear the scenario:
... to show the significance of this step of the experimental analysis, and to compare with other literature results which disregard this point.
L434
....Both measurement techniques give similar results up to the age of ca 60 days.
for similar: quote a number : within 10%? 15% refer to POINT B
for 'ca', it is typically used as 'ca.'

 Author Response

The authors would like to thank the reviewer for comments and suggestions. All changes made to the manuscript are written/displayed in red color in the revised version of the manuscript and are listed in the attached file.

Reviewer 2 Report

Some minor corrections are needed

Line 55: See chapter 2, I think section 2

Line 150: frequency of 1 min. period of 1 min or frequency of 1/min

Line 178: see fig. b, it is Fig. 1b? Because fig b not found.

Line 231: in Tab. 2. To give the the meaning of the numbers in parentheses in Table 2. It is standard deviation of measurements?

Author Response

The authors would like to thank the reviewer for comments and suggestions. All changes made to the manuscript are written/displayed in red color in the revised version of the manuscript and are listed in the attached file.

Round 2

Reviewer 1 Report

Authors have made an important effort to complete extensively the paper  with quantitative information to support the discussion in key aspects. Considering the experimental character of the paper, the new information is certainly of interest to the reader. 

The paper has gained both in the data content (tables 3-4-5, graph quality, conclusion points) and discussion content and length.  

Some points, mentioned below, still remain to be cleared up by the Authors to make a final version of the paper.     

Reviewer encourages the Authors to make the extra effort to add the needed information at the required points and provide a ready to publish paper.

POINT 1

Authors do not declare the uncertainties of the measurements done. Looking at the values in the tables, it seems rather possible that the spread of measured values (quoted as SSD, etc) are simply higher than the uncertainty of any gauge. Therefore uncertainties are not causing the measured spread in the results, and their particular values are not changing nor the discussion or the results. The statistical spread measured in actual samples is the important parameter.  

Please state that situation (if so), to avoid reader's confusion about this important point (uncertainties!). It can be at L279, after '...performed simultaneously' and before the next sentence.         

POINT 2

In the list of conclusions, authors declare that the two measuring methods are comparable to provide information up to 60 days. In figure 8, internal gauges level off, while external gauges keep a slope. Authors quote (L569) ... the differences increase due to the different positions of the gauging points.

This situation introduces an open question about what method to use after 60 days. Authors should comment on that. Either the difference is well understood and the measurements do not provide important information. Or, it is understood that the discussion of the process is considered important within a range of 60 days, and no other information is needed beyond that time. Authors should declare what is the case, and then, certainly both methods can be compared. Otherwise, the two methods can only be declared compatible for studies in that time range (60 days) up to a certain spread of values, but not beyond that time window.   This situation has to be addressed by Authors.      

Please consider also the following points as important improvements:

L192-ff
Certainly K.Hollan deserves all merits for the gauge, but the reader deserves a simple description of the measurement method (even more considering that there are no other references). Based on figures -1 and fig4-b the reviewer supposes that Hollan's gauge is a kind of transducer using the 'knobs' (embedded markers) set in the specimens to evaluate the strain by using strain-gauges.  A simple statement will help the reader.
caption Figure-7
a re-formulation of strain and temperature measurements may help:
Process of early-age shrinkage recorded by inductive sensor of shrinkage moulds (IS_ShM) and embedded strain gauge (W-SG_ShM) of specimens cured under dry air (black lines) and autogenous (grey lines) conditions. The recorded internal temperature (Tin) of specimens (colour lines) is also displayed.  

TABLE-4
the entry  W-SG_ShM-100_D-A
is possibly (!?) 
W-SG_ShM-100
since D-A or AU is analyses below
L 593-ff, about  costs: 

The discussion should be at the end of Section-5, and only indicated on the conclusions Section.

The cost analysis made by authors is important. However it seems too local in time and place to be a reference for a general reader in worldwide markets.

Maybe the authors can make a more abstract assessment, kind of:  

... The embedded strain gauges (wire type) are single use units. Additionally, the cost and the availability may be very depending on commercial conditions at each country. In the case of shrinkage moulds the costs of the measurement gauges can be reduced by more than 50 %, and the equipment is re-usable. However, the main disadvantage is the high cost of the shrinkage moulds.  In this test experience, the overall cost of one test unit of shrinkage moulds costs about 36 times the test unit with embedded strain gauges. However, considering the simpler setting of the gauges, the associated measuring equipment and the large number of tests associated to batch-test procedures in testing activities, the use of shrinkage moulds clearly results a highly competitive method in the pursue for cost-effective HPC techniques.       

CONCLUSIONS

conclusions are now a bit scattered, repeat content, and need some 'packing':

bullet #3 is redundant, as it is concluded already in #1
remove #3 (.. Both measurement techniques are suitable for the measurement of the autogenous shrinkage.)
and add at the end of #1 if considered necessary to the stress that point, something like:
... specifically for the autogenous shrinkage    

bullet #2 and #4  refer to the same information (before 60 days). merge together, removing the second line of #2 (after this time...) and the first line of #4 (It is shown...).    

bullet #5, comparing the methods. it could include a line about the demonstrated cost competitiveness of the mould method, to summarize the previous discussion in section 5.     

bullet #6 the message seems clear, but the differences are found when comparing what ? the values of the whole process, including the plastic stage ? specify in the second sentence how the difference arises, otherwise the message is loose.    

And finally, remove from the conclusions the last paragraph, moving the economic aspect into the previous section (as mentioned above).      

Author Response

The authors would like to thank the reviewer for comments and suggestions. All changes made to the manuscript in the second round of revisions are written in blue color in the revised version of the manuscript and are listed in the attached file.
